# Neuroprotective Effects of *Coreopsis lanceolata* Flower Extract against Oxidative Stress-Induced Apoptosis in Neuronal Cells and Mice

**DOI:** 10.3390/antiox10060951

**Published:** 2021-06-12

**Authors:** Hyung Don Kim, Ji Yeon Lee, Jeong-Yong Park, Dong Hwi Kim, Min Hye Kang, Hyun-A Seong, Kyung Hye Seo, Yun-Jeong Ji

**Affiliations:** 1Department of Herbal Crop Research, National Institute of Horticultural & Herbal Science, Eumsung 27709, Korea; khd0303@rda.go.kr (H.D.K.); kimodh@korea.kr (D.H.K.); mohak2@korea.kr (M.H.K.); 2Department of Biochemistry, School of Life Sciences, Chungbuk National University, Cheongju 28644, Korea; haseung@cbnu.ac.kr; 3MEDIOGEN, Co., Ltd., Jecheon 27159, Korea; ljy341@naver.com; 4Department of Medicinal Plant Resources, Andong National University, Andong 36729, Korea; yong0433@cbnu.ac.kr; 5Development of Horticultural Crop Research, National Institute of Horticultural & Herbal Science, RDA, Jeonju 55365, Korea

**Keywords:** *Coreopsis lanceolate*, apoptosis, MPTP, neuroprotective, PC12 cells

## Abstract

*Coreopsis lanceolata* L. is a perennial plant of the family Asteraceae, and its flower is known to contain flavonoids with various bioactivities. We evaluated the effect of *Coreopsis lanceolata* L. flower (CLF) extracts on H_2_O_2_-induced oxidative stress (OS) in neuronal cells and mouse neurons. The flowering part of CL was used as CLF1 (70% ethanol extract) and CLF2 (water extract), and 10 types of phenolic compounds were quantified using high-performance liquid chromatography. To evaluate the neuroprotective effects of CLF, the antioxidant activities of the extracts were measured, and the expression levels of antioxidant enzymes and proteins related to OS-induced apoptosis in neuronal cells and mouse neurons treated with the extracts were investigated. In the in vitro study, CLF ameliorated H_2_O_2_-induced oxidative stress and induced the expression of antioxidant enzymes in PC12 cells. Furthermore, CLF1 enhanced the expression of the Bcl-xL protein but reduced the expression of Bax and the cleavage of caspase-3. In the same manner, CLF1 showed neuroprotective effects against OS in vivo. Pretreatment with CLF1 (200 mg/kg) increased the Bcl-2 protein and decreased Bax compared with the 1-methyl-4-phenylpyridinium ion (MPP+)-treated C57BL/6 mice model group. Our results suggest that the protective effects of CLF1 on MPP+-induced apoptosis may be due to its anti-apoptotic activity, through regulating the expression of the Bcl-2 family. CLF1 exerts neuroprotective effects against OS-induced apoptosis in PC12 cells in a Parkinson’s disease model mouse. This effect may be attributable to the upregulation of Bcl-2 protein expression, downregulation of Bax expression, and inhibition of caspase-3 activation. These data indicate that CLF may provide therapeutic value for the treatment of progressive neurodegenerative diseases.

## 1. Introduction

Most human neurodegenerative diseases such as Alzheimer’s disease (AD) and Parkinson’s disease (PD) are related to biological aging processes [1,2]. One of the aging processes is to stop the accumulation of and increase in oxidative damage. This has also been linked to the prevention of neurodegenerative diseases. The increase in oxidative stress (OS), which reduces the activity of antioxidant enzymes, occurs when normal cells in the body are attacked by reactive oxygen species (ROS), including superoxides and hydrogen peroxide (H_2_O_2_) [3]. Antioxidant mechanisms are associated with decreased intracellular production of ROS and malondialdehyde, increased glutathione content, and increased antioxidant activity of superoxide dismutase (SOD), catalase (CAT), and glutathione peroxidase (GPx) [4]. By inhibiting the increase in OS and the accumulation of oxidative damage, these antioxidant enzymes play essential roles in the prevention of AD and PD due to lipid peroxidation in the brain.

In a recent study, it was reported that compounds isolated from edible flowers have high neuroprotective effects in various cell and animal models [5,6]. Kwon et al. [6] revealed that phenolic compounds isolated from edible flower extracts exerted neuroprotective effects in glutamate-treated HT22 cells.

*Coreopsis lanceolate* (CL) is a perennial grass belonging to the family Asteraceae (Compositae) and contains various phenolic compounds. It is widely distributed mainly in the Americas and East Asia. In Korea, yellow flowers bloom in May–July [7]. The phytochemical constituents of CL flowers (CLF) have been extensively studied. CLF contain acetylene compounds, phenolic compounds, and flavonoids (aurones, chalcones, flavanones, and flavanols) [8]. Previous in vitro and in vivo study data have demonstrated that extracts and constituents of CLF exhibit diverse biological activities and anti-diabetic [9], anti-cancer [10], anti-inflammatory [11], anti-allergenic [12], nematicidal [13], antileukemic [14], and antioxidant [15] effects. Phenols and flavonoids derived from CLF showed protective effects against radical-induced oxidative damage and attenuated toxicity in Hep G2, Caco-2, RAW264.7, and PC-12 cells [11].

Recently, Kim et al. isolated novel anti-inflammatory flavanones from CLF and confirmed their suppression of ROS, nitric oxide (NO), inducible nitric oxide synthase (iNOS), and COX-2 [11]. However, the neuroprotective effects against OS, and the corresponding mechanisms of CLF in vitro and in vivo have not yet been reported.

In the present study, we investigated the protective effects of CLF extract against neuronal damage under conditions of H_2_O_2_-induced cell injury in vitro and in vivo.

We also investigated the protective mechanisms of CLF extract against neuronal damage via the regulation of antioxidant enzymes (SOD and CAT) and apoptotic signaling.

## 2. Materials and Methods

### 2.1. Chemicals, Antibodies, and Instruments

PC12 neurons were purchased from ATCC (Manassas, Virginia, USA) and used for experiments. One hundred units/milliliter of penicillin/streptomycin, 10% fetal bovine serum (FBS), Dulbecco’s modified Eagle’s medium (DMEM), Ethylenediaminetetraacetic acid (EDTA) reagent were purchased from Sigma Aldrich (St. Louis, MO, USA). 3-(4,5-dimethylthiazol-2-yl)-5-(3-carboxymethoxyphenyl)-2-(4-sulfophenyl)-2H-tetrazolium (MTS), which was used to measure cell viability and ROS scavenging ability, was from Bio Basic (Markham, Canada), dimethyl sulfoxide (DMSO) was from Bio Pure (Burlington, ON, Canada), and dichlorofluorescein diacetate (DCF-DA) was from Sigma-Aldrich (St. Louis, MO, USA). 1-methyl-4-phenyl-1,2,4,5-tetrahydropyridine (MPTP) was purchased from TCI Co. Ltd. (Tokyo, Japan). RIPA cell lysis buffer was purchased from GenDepot (Katy, TX, USA). The fluorescence microscope was purchased from Carl Zeiss (Oberkochen, Germany). Bradford and Enhanced Chemiluminescence (ECL) reagents for protein analysis were purchased from Bio-Rad (Hercules, CA, USA). Primary antibodies anti-Bax (1:1000), anti-Bcl-2/Bcl-xL (1:1000), anti-caspase-8,9,3 (1:1000), anti-SOD (1:1000), and anti-Catalase (1:1000) were purchased from Abcam (Cambridge, UK). β-actin and secondary antibody anti-mouse (1:2000) and anti-rabbit (1:2000) were purchased from Santa Cruz Biotechnology (Dallas, TX, USA). For ChemiDoc image analysis, a product made by Bio-Rad (CA, USA) was used. Material analysis was performed using a high-performance liquid chromatography (HPLC) 2790/5 system equipped with a photodiode array (PDA; 2996; Waters, Milford, MA, USA). HPLC solvents acetonitrile and water were purchased from Fisher Scientific Ltd. Distillation was performed using a vacuum reflux evaporator system (BÜCHI, Sankt Gallen, Switzerland).

### 2.2. Preparation of Plant Materials

CLF were collected from Eumsung (Chungcheongbuk-do, Korea) in 2018 (Figure 1). CLF were freeze dried for 1 week (20 mTorr, −40 °C) and then collected. For the preparation of the extract, CLF (100 g each) were crushed and sieved through a test specimen (aperture 1.40 mm, wire 0.71 mm) to add 70% ethanol (CLF1) and water (CLF2) in a ratio of 1:10 for 24 h at room temperature and extracted 3 times. After filtration, the extract was evaporated, freeze dried (20 mTorr, −40 °C) for 1 week, and stored in a −80 °C deep freezer.

### 2.3. Analysis of Antioxidant Components

The total phenolic content (TPC) of the extracts was calculated using the Folin–Ciocalteu reagent method [16]. We added 500 μL of the sample or standard, diluted with 50 μL Folin–Ciocalteau reagent (1 N; Sigma-Aldrich), and left it to stand for 3 min. Next, 100 μL of 20% Na_2_CO_3_ was added to the mixture, which was then shaken and incubated at room temperature for 60 min. For the analysis of the total phenol content, Folin–Ciocalteau reagent, which colored blue when reacted with phosphomolybdic acid, was used, and the value was expressed as gallic acid equivalent (mg GAE/g). All samples were analyzed in 3 repetitions, and absorbance was measured at 732 nm using a multi-plate reader (BioTek Instruments, Winooski, VT, USA). The total flavonoid content (TFC) in the extracts was determined using the Folin–Ciocalteu reagent method [16], with some modifications. A total of 150 μL of the solution (standard or CLF1 and CLF2) was mixed with 0.3 mL of NaNO_2_ (5%, *w*/*v*). After 5 min, 10 μL of AlCl_3_ (10%, *w*/*v*) was added. The sample was mixed and neutralized after 6 min with 0.5 mL of 1 M NaOH solution. The mixture was then left for 10 min at room temperature. The catechin equivalent (mg of CAE/g) was the TFC standard, and absorbance was detected at 510 nm using a multi-plate reader (BioTek Instruments).

### 2.4. HPLC Analysis of Antioxidant Compounds

#### 2.4.1. Sample Preparation for HPLC

HPLC analysis (Agilent 1200 series; Agilent Technologies, Santa Clara, CA, USA) of the phenol-rich fraction was conducted according to Kim et al. [17]. CLF1 and CLF2 (5 g) were redissolved in water and then fractionated with a mixture of ethyl acetate and ether (1:1 = *v*/*v*). Each fraction was dissolved in methanol (10 mg/mL each) following concentration under reduced pressure.

#### 2.4.2. HPLC Analysis of Phenolic Compounds

We applied the modified gradient conditions for the HPLC analysis according to Kim et al. [17]. For qualitative and quantitative analysis, a reverse phase system with synergistic fusion RP column (250 × 4.6 mm, 4 μm; Phenomenex, Torrance, CA, USA) was used at 35 °C. The mobile phase consisted of 0.5% acetic acid in water and acetonitrile. The standard solutions were prepared at a constant volume with methanol and water. The elution program was as follows (B%): 2%, 5 min; 2–5%, 12 min; 5–8%, 17 min; 8–30%, 65 min; 30%, 68 min; 50%, 78 min; 100%, 100 min. The injection volume, flow rate, and wavelength were 10 μL, 1.0 mL/min, and UV 280 nm, respectively. Gallic acid, chlorogenic acid, homogentisic acid, caffeic acid, catechin, ferulic acid, *ρ*-coumaric acid, naringin, quercetin, and cinnamic acid were used as standards (Sigma-Aldrich). All solutions were filtered through a polyvinylidene difluoride (PVDF, 0.22-μm) membrane (Pall Co., Port Washington, NY, USA).

### 2.5. Antioxidant Assay

#### 2.5.1. ABTS^+^ Radical Scavenging Assay

ABTS + (2,2′-azino-bis 3-ethyl benzothiazolin-6-sulfonic acid) radical scavenging activity was measured by slightly modifying the method of Pourmorad F et al. [18]. For each antioxidant assay, ascorbic acid (AA) solution (-) was used as a standard. Assay results were obtained using a multi-plate reader (BioTek Instruments) set at wavelengths appropriate for each assay. All samples were analyzed in 3 repetitions. The ABTS scavenging assay was evaluated by the modified method. An ABTS tablet (Sigma-Aldrich) with 2.6 mM potassium persulfate was dissolved in distilled water, and the ABTS solution was prepared 12–16 h in advance. After dilution in distilled water to an absorbance of 0.700 ± 0.01 at 734 nm, 20 μL sample was mixed with 180 μL of the ABTS solution and reacted in the dark for 30 min. The absorbance value was measured at 732 nm. The half-maximal inhibitory concentration (IC50) of the ABTS radical was used in the sample.

#### 2.5.2. DPPH Radical Scavenging Assay

Measuring DPPH radical scavenging activity was conducted by modifying the method of Brand-Williams et al. [19]. To 3 mL of DPPH (40 mg/L) dissolved in 100% methanol, 20 µL of standard and extract was added. When the absorbance of the blank was 0.700 ± 0.01, the scavenging ability of the samples was determined at 515 nm. The IC_50_ was used for samples.

### 2.6. Cell Culture

The PC12 cell line was used with Dulbecco’s modified Eagle’s medium (DMEM; Gibco, Waltham, MA, USA) containing 10% fetal bovine serum (FBS; Gibco) and 1% penicillin/streptomycin (Gibco) in a 5% CO₂ incubator maintaining a temperature of 37 °C. It was used in the experiment while culturing at passages 5 through 10. The medium was changed every 2 days for subculture. Undifferentiated cells were used at passages 5–10 in all experiments.

### 2.7. Cell Viability Assay

To evaluate the effects of the CLF extract on cell viability, cell viability was measured with a CellTiter 96 Aqueous One Solution Cell Proliferation Assay Kit (Promega Corporation, Madison, WI, USA). After culturing by dispensing cells at a concentration of 1.0 × 105 cells/mL in a 96-well plate, the CLA extract was treated with each concentration (50, 100, 200 μg/mL) for 24 h. After 24 h, MTS 3-(4,5-dimethylthiazol-2-yl)-5-(3-carboxymethoxyphenyl)-2(4-sulfophenyl)-2H-tetrazolium, inner salt solution was added to each well. After culturing by dispensing cells at a concentration of 1.0 × 105 cells/mL in a 96-well plate, CLA extracts were treated at each concentration (50, 100, 200 μg/mL) for 24 h. After 24 h, the MTS [3-(4,5-dimethylthiazol-2-yl)-5-(3-carboxymethoxyphenyl)-2(4-sulfophenyl)-2H-tetrazolium, inner salt] solution was added to each well for 1 h. The plate was shaken for 5 min at room temperature to dissolve the precipitated formazan. The absorbance was measured at 490 nm using a Synergy H1 multi-plate reader (BioTek Instruments, Inc., Winooski, VT, USA).

### 2.8. Generation of Intracellular ROS

To confirm intracellular ROS production, PC12 cells seeded in 96-well plates (1 × 105 cells/mL) were treated with extracts at concentrations of 50, 100, and 200 μg/mL or vehicle control (DMSO, in a final concentration of 0.2%) for 24 h, followed by H_2_O_2_ (100 μM) for 30 min. After diluting 20 μM DCFH-DA in serum-free media, it was dispensed onto a plate, and the cells were reacted in a dark room at 37 °C for 30 min.

After washing three times with acid-buffered saline (PBS; Gibco), the images were observed with a fluorescence microscope (Carl Zeiss) to evaluate intracellular ROS production. Fluorescent DCF was measured at 485 nm and 535 nm using a multi-plate reader (BioTek Instruments) and expressed as a numerical value.

### 2.9. Animals and Treatments

All animal experiments were approved by the Institutional Animal Care and Use Committee at the National Institute of Horticultural and Herbal Science, Jeonju, Korea (IACUC approval number: NIHHS-2018-001) for the in vivo study. Six-week-old male C57BL/6 mice were used in this study (Central Lab. Animal Inc., Seoul, Korea). The mice were housed in standard polypropylene cages (four mice per cage) at a controlled temperature of 25 ± 2 °C and humidity of 40–60% under a 12-h light/dark cycle and given free access to standard rodent chow (Deahan Biolink Co., Eumseong, Korea) and water. All solutions used for the treatments were freshly dissolved on the experimental days in physiological saline (0.9% NaCl). After a week of acclimatization, the mice were randomly divided into five groups (n = 8 per group), as follows: normal group (0.9% NaCl), MPTP-treated group, MPTP + CLF1 group (50 mg/kg), MPTP + CLF1 group (100 mg/kg), and MPTP + CLF1 group (200 mg/kg). The extracts were administered orally at a dose of 50–200 mg/kg/day for 16 days with a feeding needle catheter, and the mice received an intraperitoneal (i.p.) injection of 200 μL MPTP (20 mg/kg/day) for the last 10 days. The normal group received oral subcutaneous and i.p. injections of the same volume of 0.9% NaCl. The mice were randomly divided into five groups (n = 8 per group), as follows: normal group (0.9% NaCl), MPTP-treated group, MPTP + CLF1 group (50 mg/kg), MPTP + CLF1 group (100 mg/kg), and MPTP + CLF1 group (200 mg/kg). The complete experimental schedule is shown in Figure 2. Following euthanasia, the mouse brains were dissected and prepared for Western blot analysis. The brain tissues were rapidly frozen on ice and kept at −80 °C until use.

### 2.10. Preparation of Protein Samples

PC12 cells and mouse brain tissues were collected and lysed in RIPA buffer (GenDEPOT, Katy, TX, USA) with 1× protease and phosphatase inhibitor cocktail (GenDEPOT) for 30 min and then centrifuged at 12,000 rpm for 30 min at 4 °C.

Mitochondrial and cytosolic proteins in the PC12 cells were assayed with a Mitochondria/Cytosol Fractionation Kit (Abcam). All protein contents were measured by the Bradford protein assay (Bio-Rad).

### 2.11. Western Blot Immunoassay

After harvesting PC12 cells to confirm protein expression, RIPA buffer was added and reacted at 4° C, for 1 h, followed by centrifugation at 12,000 rpm for 30 min to separate the supernatant protein.

The separated protein was quantified using the Bradford protein assay kit, a protein quantification reagent used in the experiment. The same sample was separated by electrophoresis using 10% sodium dodecyl sulfate (SDS)-polyacrylamide gel. The sample was transferred to a polyvinylidene difluoride (PVDF) membrane, blocked for 1 h, treated with the primary antibody, and reacted overnight at 4 °C. After reacting with the secondary antibody for 1 h at room temperature, it was washed with PBS-T and reacted with enhanced chemiluminescence (ECL) solution to confirm the expression of a specific protein using the hemiDoc Imaging System (Bio-Rad).

### 2.12. Statistical Analysis

All experimental results were expressed as mean ± standard deviation (means ± SD), and each was analyzed by analysis of variance (ANOVA) using the statistical program Statistical Package for the Social Sciences (version 21.0, SPSS Inc., Chicago, IL, USA). The difference between the groups was tested for the significance of each group at the *p* < 0.05 level using Tukey’s test. Analyses were performed using GraphPad Prism (ver. 5.02; GraphPad Software Inc., San Diego, CA, USA).

## 3. Results

### 3.1. Phenolic Compounds Contents of CLF1 and CLF2 Identified by HPLC

Phenolic acids (PAs), which are antioxidant components of CLF1 and CLF2, were analyzed by a Waters HPLC (ACME 9000 system, Younglin, Anyang, Korea) equipped with a PDA detector. Phenol-rich fractions were obtained from CLF1 and CLF2 to determine the phenolic compounds. The chromatograms of the phenolic compounds and phenol-rich fractions of CLF1 and CLF2 are shown in Figure 3. The contents of homogentisic acid (1.22-fold), chlorogenic acid (2.54-fold), (+)-catechin (1.62-fold), caffeic acid (1.53-fold), and naringin (2.44-fold) were higher in CLF1 than CLF2. Conversely, the contents of ferulic acid (3.60-fold), quercetin (2.92-fold), and cinnamic acid (1.36-fold) were higher in CLF2 than CLF1. CLF1 and CLF2 might have higher scavenging effects because of these phenolic compounds (Table 1).

### 3.2. Antioxidant Components and Activities of CLF1 and CLF2

In this experiment, the phenolic extract yields of CLF1 and CLF2 were 25.89% and 16.63%, respectively. The phenolic extract yield of 70% ethanol was more than 1.5 times higher than for the water extract, possibly due to its high solubility. The total phenolic content (TPC) and total flavonoid content (TFC) in edible flowers are positively correlated with high antioxidant activity [20,21]. Table 2 shows the total polyphenol and flavonoid contents of CLF1 and CLF2. The total phenol content in CLF1 and CLF2 was 13.7 mg GAE/g and 14.8 mg GAE/g, respectively, with no difference in content. Flavonoid contents were also 13.0 mg CAE/g and 18.1 mg CAE/g, respectively, showing similar contents. According to the extraction method, the extraction yields of CLF1 and CLF2 were 25% and 16%, respectively, and there was no significant difference between the two solvents. To investigate the antioxidant effects of CLF1 and CLF2, DPPH radical scavenging activity and ABTS+ radical scavenging activity were analyzed using ascorbic acid (AA) as a positive control. As a result, IC50 values were 111.8 and 176.9 μg/mL for ABTS+ and 24.5 and 39.9 μg/mL for DPPH. CLF1 showed a higher radical scavenging ability.

### 3.3. Effects of CLF1 and CLF2 on H_2_O_2_-Induced ROS Generation in PC12 Cells

To determine whether CLF had any effects on cell viability, PC12 cells were treated with 50–200 μM CLF for 24 h. The results indicate that CLF extract had no significant effect on cell viability (Figure 4a). Therefore, CLF concentrations up to 200 µg/mL were used in the subsequent experiments. The effects of CLF on H_2_O_2_-induced ROS generation in PC12 cells were analyzed by staining with DCFH-DA and detecting and quantitatively measuring the fluorescence intensity, which was proportional to the amount of ROS generated, by confocal microscopy and a multi-plate reader. As shown in Figure 4c, H_2_O_2_ increased intracellular ROS levels, and CLF1 and CLF2 reduced the levels of H_2_O_2_-induced ROS in a dose-dependent manner in PC12 cells. Quantitative evaluation of intracellular ROS generation showed that ROS levels increased approximately 1.3-fold in the H_2_O_2_-treated group compared to the control but were reduced in a dose-dependent manner (50, 100, and 200 μg/mL) when treated further with CLF1 and CLF2 (CLF1, 65.25, 63.63, and 57.50% of the H_2_O_2_-treated group; CLF2, 81.99, 75.02, and 74.58% of the H_2_O_2_-treated group; Figure 4d). CLF1 inhibited H_2_O_2_-induced ROS generation more efficiently than CLF2. We investigated the correlation between the antioxidant activity of CLFs and the content of phenolic compounds (chlorogenic acid, quercetin, caffeic acid, and cinnamic acid) (Table 3). IC50s of ABTS+ and DPPH scavenging activities were inversely correlated with caffeic acid (−0.996, −0.900), chlorogenic acid (−0.994, −0.893), quercetin (−0.994, −0.890), and cinnamic acid (−0.976, −0.855). Additionally, ROS in PC12 cells were inversely correlated with chlorogenic acid (−0.914), caffeic acid (−0.913), quercetin (−0.906), and cinnamic acid (−0.842). These results show that the phenolic compounds in CLFs increase radical and cellular ROS scavenging activities. Therefore, CLF1 apparently inhibited H_2_O_2_-induced ROS generation in PC12 cells more efficiently than CLF2 because CLF1 is richer in antioxidants such as chlorogenic acid and caffeic acid than CLF2. Phenolic compounds in CLFs are expected to inhibit ROS production via inducing endogenous antioxidant enzymatic activities.

### 3.4. Effects of CLF1 on Oxidative Stress-Related Proteins in PC12 Cells

The antioxidant protein levels of SOD2 and CAT inversely correlated with levels of ROS. CLF1 efficiently reduced H_2_O_2_-induced ROS generation in PC12 cells. Therefore, to explore the ability of CLF1 to inhibit OS, we determined the expression of antioxidant enzymes such as SOD2 and CAT (Figure 5). Treatment of CLF1 in PC12 cells dramatically increased the expression of antioxidant enzymes. Our data suggest that CLF1 can reduce H_2_O_2_-induced OS by upregulating the expression of antioxidant enzymes in cells.

### 3.5. Effects of CLF1 on the Mitochondrial Apoptotic Pathway in H_2_O_2_-Treated PC12 Cells

Mitochondrial-related apoptosis proteins are associated with H_2_O_2_-induced neuronal cell loss. In this study, the expression of Bax and Bcl-xL was confirmed on the mitochondrial membranes. Bax and Bcl-xL are involved in apoptosis. Treatment of cells with CLF1 at concentrations of 50, 100, and 200 μM decreased Bax expression, upregulated Bcl-2 expression, and reduced the Bax/Bcl-xL ratio at 100 μM concentrations. It was confirmed that CLF1 has a neuroprotective effect (Figure 6a). In brain nerve cell damage, increased cytochrome c expression due to Bax expression activates caspase-9, a caspase family member in the form of a cleaved protein, which activates caspase-3 and causes apoptosis [22]. Activation and cleavage of caspase-3 was confirmed as the mechanism underlying the protective effect against apoptosis (Figure 6b). Activation of caspase-3 was higher in the H_2_O_2_-only group, but CLF1 treatment significantly inhibited this activation in a dose-dependent manner, confirming the protective effect of CLF1 on apoptosis.

### 3.6. Effects of CLF1 on Oxidative Stress-Related Proteins In Vivo

To investigate the neuroprotective effects of CLF1 against OS in vivo, OS was induced with MPTP in mice. Male C57BL/6 mice were orally treated with CLF1 (50, 100, and 200 mg/kg) for 16 days. From day 6 onwards, MPTP (30 mg/kg) was injected i.p. for the last 10 days. The results demonstrate that CLF1 could improve SOD 1 and CAT activity and reduce OS. In MPTP-induced mouse brains, administration of CLF1 significantly increased the expression of SOD 1 and CAT in a dose-dependent manner (Figure 7). When CLF1 was administered orally, the protein expression was the same as in the in vitro studies, indicating that CLF1 also has an antioxidant effect in vivo.

### 3.7. Effects of CLF1 on the Apoptotic Pathway In Vivo

We evaluated the effects of CLF1 on the apoptotic pathway in MPTP-treated mouse brains. MPTP treatment increased the Bax/Bcl-xL and Bax/Bcl-2 ratios compared to the control mice; this was restored by CLF1, as seen in vitro (Figure 8a). MPTP enhanced the activation and cleavage of caspase-3 through activation and cleavage of caspase-8 and -9 in a mouse model MPTP treatment, and this was attenuated by CLF1 (Figure 8b). The results show that CLF1 could protect neurons from MPTP-induced apoptosis by downregulating the apoptotic pathway. Our study suggests that CLF1 could prevent MPTP-induced behavioral deficits, OS, apoptosis, dopaminergic neuronal degeneration, and dopamine depletion.

## 4. Discussion

OS is involved in various neurodegenerative diseases, including Alzheimer’s and PD [23]. OS plays a major role in neurodegenerative diseases via free radical attacks on nerve cells. The neurotoxicity associated with excessive ROS production contributes to protein misfolding, glial cell activation, mitochondrial dysfunction, and subsequent cellular apoptosis [24]. Under physiological conditions, the antioxidant defense system consists of endogenous antioxidant enzymes, such as CAT, SOD, and GPx, and exogenous antioxidants obtained by dietary intakes of vitamin C, vitamin E, and carotenoids [25]. Flower extracts have shown antioxidant properties in cell and animal models by increasing antioxidant enzymes (CAT, SOD, and GPx) and reducing free radicals (ROS) and peroxidation end products [26]. High antioxidant activity and free radical scavenging ability have been found in most edible flowers [27]. The potent antioxidant activity of flowers was strongly correlated with plant chemicals such as phenolic acids, flavonoids, and terpenes [20,28].

Phenolic compounds have antioxidant activities because they can terminate free radical chains and chelate redox-active metal ions; therefore, they can protect tissues against free oxygen radicals and lipid peroxidation [29]. CLF, which is currently attracting attention as a new antioxidant, is a natural material that contains various phenolic compounds such as phenylpropanoids and flavonoids. Phenolic compounds from CLF are beneficial, as they have proven anti-cancer, antibacterial, anti-inflammatory, and antioxidant effects [10,11,12,13,14,15,16].

This study explored the effects of CLF extracts on H_2_O_2_-induced PC12 cell injury and PD-like nerve damage in MPTP-treated male C57BL/6 mice. Our results show that administration of CLF extract significantly ameliorated ROS-dependent neuroapoptosis and nerve injury-induced neuropathic pain in mice.

We identified antioxidants in the phenol-rich fractions of CLF1 and CLF2 by HPLC analysis. Phenolic compounds such as caffeic acid, cinnamic acid, chlorogenic acid, and quercetin were identified. Caffeic acid and cinnamic acid, abundant in CLF1 and CLF2, have beneficial antioxidant and anti-inflammatory effects [27,30]. Caffeic acid, which is abundant in CLF1, has other beneficial effects such as anti-cancer cell proliferation activities, acts in vitro as an antioxidant, and inhibits autophagy [31,32,33,34]. According to Gao et al. [35], treatment of PC12 cells with caffeic acid induced an anti-inflammatory response and showed an ROS-inhibiting effect. Thus, caffeic acid displayed a neuronal protective effect through the reduction in OS. Caffeic acid isolated from *Erigeron annuus* leaves reduced H_2_O_2_-induced ROS in PC12 cells [36]. In addition, chlorogenic acid increases the antioxidant enzymes that reduce ROS generation, such as SOD, CAT, and GPx. Meanwhile, chlorogenic acid protected PC12 cells from oxidative damage [37]. Our results suggest that the antioxidant effects of CLF1 and CLF2 might be due to the physiological activities of phenolic compounds, such as caffeic acid, cinnamic acid, and chlorogenic acid, which contain aglycone glycoside structures. Most of the studies on the antioxidant activity of flowers are in vitro chemical-related antioxidant assays. For future studies, the focus should be on in vivo studies, and the precise mechanism of CLF extract’s antioxidant enzyme activity should be explored.

Several antioxidant enzymes, such as SOD, CAT, and GPx, are used by the body to combat OS. H_2_O_2_ is a major cause of ROS formation and is converted into highly reactive hydroxyl radicals that attack biological pathways [38]. Mitochondrial-related apoptosis proteins are associated with H_2_O_2_-induced neuronal cell loss [39]. The anti-apoptotic factor Bcl-2 inhibits the proapoptotic factor Bax, thereby promoting the release of cytochrome c [40]. Cytochrome c induces cell death by triggering the activation of caspase-3 and caspase-9 [41]. To investigate the protective effect of CLF1 and CLF2 on PC12 cell death against H_2_O_2_, we identified the underlying mechanisms by Western blotting. We showed that CLF1 could reduce H_2_O_2_-induced ROS generation by improving the expression of antioxidant enzymes, such as CAT and SOD, in vitro (Figure 5). The Bax/Bcl-xL ratio was significantly increased in cells treated with H_2_O_2_, whereas this was prevented by treatment with CLF1 (Figure 6a). To examine whether CLF1 prevented cell death by regulating the caspase cascade, we analyzed the levels of cleaved caspase-9 and -8, and the protein expression levels of cleaved caspase-3. The expression levels of cleaved caspase-9, -8, and -3 proteins were upregulated in cells exposed to H_2_O_2_ compared to non-exposed cells (Figure 6b). However, treatment with CLF1 (50, 100, and 200 μg/mL) significantly reduced the cleaved caspase-9, -8, and -3 expression induced by H_2_O_2_ in a dose-dependent manner. In particular, 200 μg/mL of CLF1 markedly inhibited cleavage of caspase-9, -8, and -3. These results suggest that CLF1 can protect against H_2_O_2_-induced PC12 cell death by downregulating the apoptotic signaling pathway.

We confirmed the neuroprotective effect of CLF1 on OS in an MPP+-induced animal PD model. Our results clearly show that CLF1 protects neurons from ROS-induced mitochondrial damage in vivo, which was induced by MPTP/1-methyl-4-phenylpyridinium ion (MPP+). MPP+ is a neurotoxin that selectively damages catecholaminergic neurons, including dopaminergic neurons, and is widely used in experimental models of PD. MPP+ can operate in conditions of extracellular and intracellular oxidation, resulting in ROS that lead to toxic molecules and thus neuronal damage. The MPP+-induced apoptosis in neuronal cells interferes with mitochondrial outer membrane permeability, which increases the release of mitochondrial cytochrome c and activation of caspase-3 [42]. Some studies reported that ROS are involved in the apoptotic mechanism of MPP+-mediated neurotoxicity and may contribute to the apoptotic processes found in PD [43]. Some antioxidants have been shown to prevent apoptotic cell death in dopaminergic cell lines, and in SH-SY5Y cells treated with MPP+ [44,45].

Our results show that CLF1 exerted its antioxidant activity by upregulating antioxidant enzymes, such as SOD and CAT, in vivo (Figure 7). CLF1 lowered the Bax/Bcl-2 ratio (Figure 8a) and reduced the activation and cleavage of caspase-9, -8, and -3 (Figure 8b). We showed that CLF1 ameliorated OS in a concentration-dependent manner and protected neurons from MPTP/MPP+-induced apoptosis by regulating the expression of apoptosis-related proteins. Our results suggest that cytoprotection of CLF1 against MPTP/MPP+-induced cell death may be associated with the attenuation of oxidative damage by inhibiting ROS generation, indicating that CLF1 exerts a neuroprotective effect by inhibiting the mitochondrial cell death pathway in vivo.

For polyphenols to exert their direct neuroprotective action, they must penetrate the blood–brain barrier (BBB). The ability of flavonoids to cross the BBB is believed to be dependent on lipophilicity, but small molecule phenols have been reported to cross the BBB via amino acid transporters [46,47]. The most common and essential low-molecular weight phenolic compounds are simple phenol derivatives (phenolic acid, caffeic acid, coumaric acid, gallic acid, etc.) and flavonoids (catechin, quercetin, cyanidin, etc.). These can originate from endogenous metabolic processes in the brain and can be absorbed into the blood. However, the only source of polyphenols in human body fluids is food. Numerous observational and interventional studies have shown that increased polyphenol dietary intake positively affects cognitive abilities in elderly subjects [48], which is a derivative of ingested plant polyphenols produced by the gut microbiota from ingested fruits and vegetables. This indicates that it can be absorbed into the bloodstream and cross the BBB to reach brain cells. This could be direct evidence that some plant phenolics can cross the human BBB. Our study showed that 20 mg/kg orally administered CLF did not show toxicity in animal models and showed neuroprotective actions in caffeic acid and chlorogenic acid, which are abundant in CLF1. Therefore, as with the compounds investigated in our study, these polyphenols may significantly contribute to the various physiological effects of dietary intake.

However, a limitation of this study is that the efficacy of delivery of CLF1 compounds through BBB and proteins’ expression in the separated regions of brain were not investigated. We will study them when we reveal the detailed pharmacological mechanisms of CLF in further study. In summary, we evaluated the effects of CLF on H_2_O_2_-induced OS in PC12 cells and MPTP-induced PD-like conditions in a mouse model. The results show that CLF, which contains phenylpropanoids and flavonoids with potent antioxidant activity, effectively inhibited nerve cell death induced by H_2_O_2_ or MPTP. We suggest that CLF could be considered as a new antioxidant material to develop novel therapeutic agents for age-related neurological disorders.

## 5. Conclusions

This study evaluated the effect of CLF1 on H_2_O_2_-induced OS in PC12 cells and MPTP-induced PD-like symptoms in C57BL/6 mice. CLF1 exerted neuroprotective effects against apoptosis by targeting oxidant and antioxidant agents, such as SOD2 and CAT. CLF1 downregulated the mitochondrial apoptotic pathway, and the activation and cleavage of caspase-9, -8, and -3. The results suggest that CLF1 effectively inhibited OS-induced neuronal damage through antioxidant and anti-apoptotic pathways, both in vitro and in vivo. Therefore, CLF is a functional natural material that prevents neurodegenerative diseases such as PD by reducing nerve damage caused by OS.

## Figures and Tables

**Figure 1 antioxidants-10-00951-f001:**
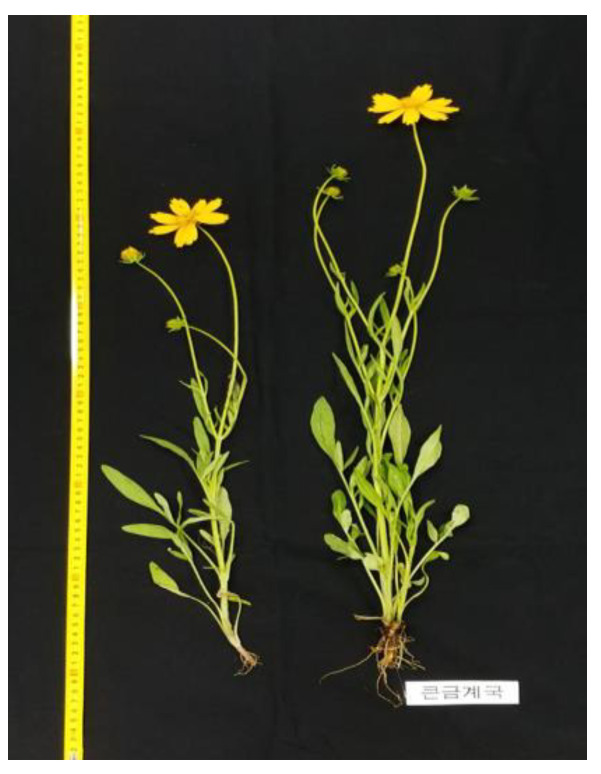
*Coreopsis lanceolate L* from Eumsung (Chungcheongbuk-do, Korea).

**Figure 2 antioxidants-10-00951-f002:**
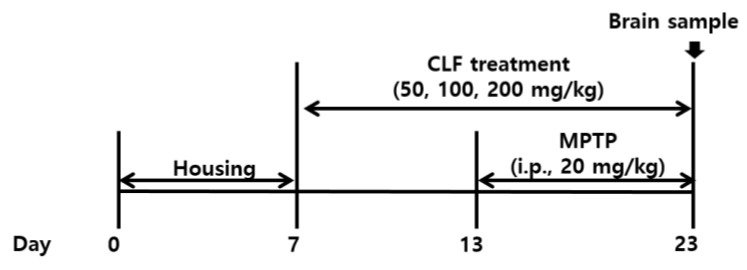
Effect of extract on body weight and food intake. Control group; MPTP-treated group (20 mg/kg, 1/day for 10 days); MPTP + CLF-treated group (50, 100, 200 mg/kg).

**Figure 3 antioxidants-10-00951-f003:**
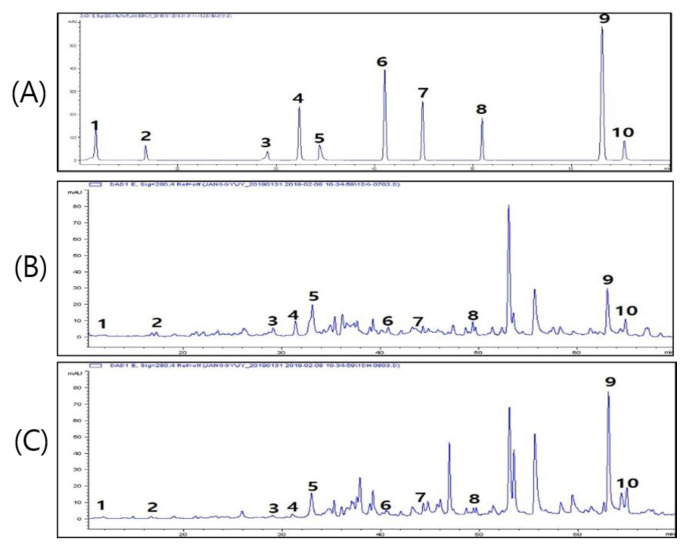
Chromatograms of the phenolic compounds from CLF1 and CLF2 based on HPLC analysis. Peak identification: 1: gallic acid, 2: homogentisic acid, 3: chlorogenic acid, 4: (+)-catechin, 5: caffeic acid, 6: p-coumaric acid, 7: ferulic acid, 8: naringin, 9: quercetin, 10: cinnamic acid. (**A**): Chromatogram of standard solution used for phenolic compound analysis (100 μg/mL each). (**B**) Chromatogram of CLF1 (10 mg/mL). (**C**) Chromatogram of CLF2 (10 mg/mL). CLF1: *C. lanceolate* flowers 70% ethanol extract; CLF2: *C. lanceolate* flowers water extract.

**Figure 4 antioxidants-10-00951-f004:**
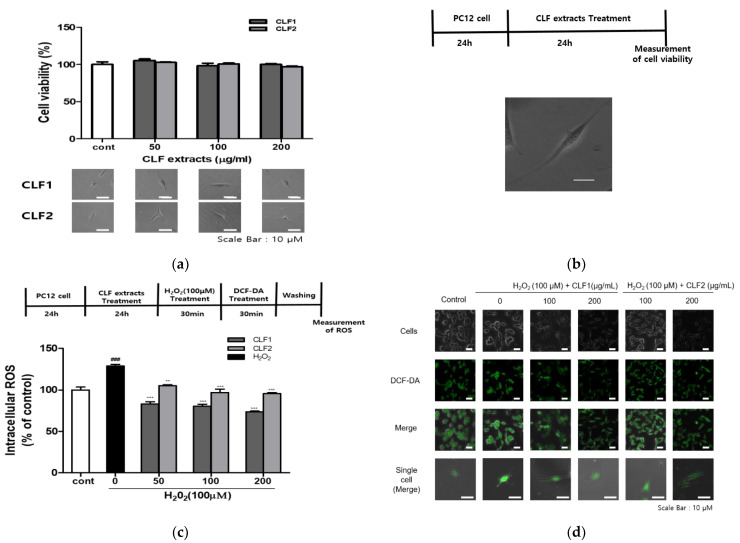
Inhibitory effect of CLF on H_2_O_2_-induced oxidative stress in PC12 cells. (**a**) PC12 cells were treated with CLF1 (50, 100, and 200 μg/mL) and vehicle control (0.2% DMSO) for 24 h. (**b**) The cells were stained with DCFDA. Then, the cells were visualized using a fluorescence microscope (scale bar; 100 μm). Representative photographs are presented. (**c**) The cells were treated with CLF1 and CLF2 (50, 100, and 200 μg/mL) or vehicle control (0.2% DMSO) for 24 h and then stimulated with H_2_O_2_ (100 μM) for 30 min. (**d**) ROS generation in PC12 cells was detected by fluorescence microscopy and a fluorescence microplate reader. Results are the mean ± standard deviation of three independent measurements. Significance was determined by one-way analysis of variance; ** *p* < 0.01, *** *p* < 0.001 vs. H_2_O_2_-induced control; ### *p* < 0.001 vs. non-treated control according to Tukey’s multiple comparison test.

**Figure 5 antioxidants-10-00951-f005:**
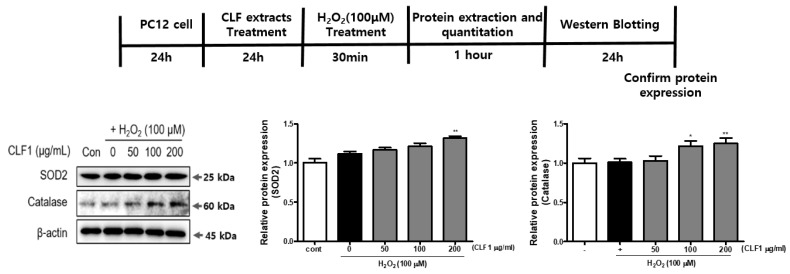
Effects of CLF1 on the expression of SOD2 and CAT in PC12 cells. Cells were treated with CLF1 (50, 100, and 200 μg/mL) or vehicle control (0.2% DMSO) for 24 h followed by H_2_O_2_ (100 μM) for 30 min. The cell lysates were then subjected to Western blot analysis. β-actin was used as a loading control. The levels of SOD2 and CAT were quantified by densitometric analysis. Significance was determined by one-way analysis of variance; * *p* < 0.1, ** *p* < 0.01 vs. H_2_O_2_-induced control; *p* < 0.001 vs. non-treated control according to Tukey’s multiple comparison test.

**Figure 6 antioxidants-10-00951-f006:**
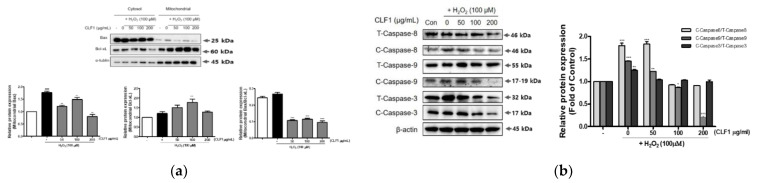
CLF1 protects PC12 cells from H_2_O_2_-induced oxidative apoptosis. Bax/Bcl-xL, Bax/Bcl-2 (**a**) and caspase 3, 9, and 8 (**b**) protein expression in the mitochondria. The cells were treated with CLF1 (50, 100, and 200 μg/mL) or vehicle control (0.2% DMSO) for 24 h followed by H_2_O_2_ (100 μM) for 30 min. Cell lysates were subjected to gel electrophoresis, and then Western blot analysis was performed. β-actin was used as a loading control. Significance was determined by one-way analysis of variance; * *p* < 0.1, ** *p* < 0.01, *** *p* < 0.001 vs. H_2_O_2_-induced control; *p* < 0.001 vs. non-treated control according to Tukey’s multiple comparison test.

**Figure 7 antioxidants-10-00951-f007:**
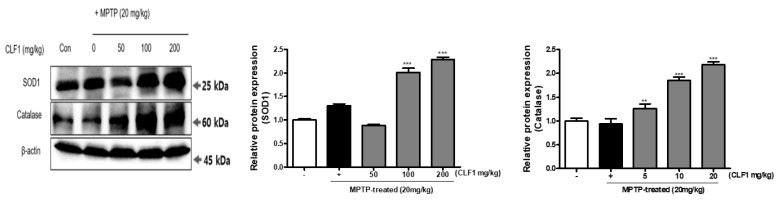
Effects of CLF1 on the protein expression of antioxidative enzymes in MPTP-treated mouse brain showing the expression levels of SOD2 and CAT. Cell lysates were subjected to gel electrophoresis, and then Western blot analysis was performed. β-actin was used as a loading control. Significance was determined by one-way analysis of variance; ** *p* < 0.01 *** *p* < 0.001 vs. MPTP-treated control; *p* < 0.001 vs. non-treated control according to Tukey’s multiple comparison test.

**Figure 8 antioxidants-10-00951-f008:**
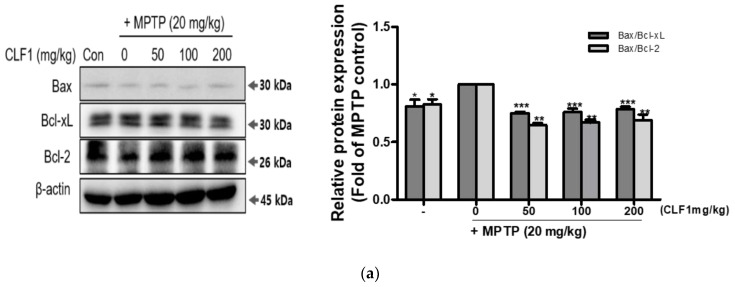
Suppression of CLF1 on the protein expression of apoptosis in MPP+-treated mouse brain. (**a**) Protein expression related Table 2. (**b**) Protein expression related to the extrinsic pathway (cleaved (C), pro-caspase-8, -9, and -3). All protein expression levels were quantified by normalizing to β-actin levels, and their fold changes relative to the control were calculated. Significance was determined by one-way analysis of variance; * *p* < 0.05, ** *p* < 0.01, *** *p* < 0.001 vs. MPTP-treated control; *p* < 0.001 vs. non-treated control according to Tukey’s multiple comparison test.

**Table 1 antioxidants-10-00951-t001:** Phenolic compound contents of CLF1 and CLF2 identified by HPLC (n = 3).

Phenolics	Contents (mg/g extract d.b.)	
	CLF1	CLF2
Gallic acid	8.45 ± 0.00	8.794 ± 0.00
Homogentisic acid	5.62 ± 0.04	4.57 ± 0.19
Chlorogenic acid	15.44 ± 0.03	6.07 ± 0.134
(+)-Catechin	7.80 ± 0.05	4.79 ± 0.04
Caffeic acid	43.29 ± 0.10	28.45 ± 0.263
p-Coumaric acid	0.56 ± 0.01	N.D.
Ferulic acid	1.25 ± 0.01	4.50 ± 0.22
Naringin	4.04 ± 0.16	1.65 ± 0.362
Quercetin	3.58 ± 0.01	10.46 ± 0.128
Cinnamic acid	18.31 ± 0.64	25.05 ± 0.081

CLF1: *C. lanceolate* flowers 70% ethanol extract; CLF2: *C. lanceolate* flowers water extract. N.D.: not detected. All values are mean ± SD.

**Table 2 antioxidants-10-00951-t002:** Yield and total polyphenol and flavonoid contents of CLF1 and CLF2.

Sample	Total Polyphenol (mg GAE/g)	Total Flavonoid (mg GAE/g)	ABTS+ (IC_50_, μg/mL)	DPPH (IC_50_, μg/mL)	Yields (%)
CLF1	13.7 ± 0.2 ^a^	14.8 ± 1.5 ^b^	24.5 ± 0.4 ^b^	111.8 ± 4.5 ^b^	25.89 ± 0.3 ^b^
CLF2	13.0 ± 0.1 ^b^	18.1 ± 1.3 ^a^	39.9 ± 1.4 ^b^	183.3 ± 1.7 ^a^	16.63 ± 0.2 ^a^
AA	-	-	5.9 ± 0.1 ^c^	4.7 ± 0.3 ^c^	-

CLF1: *C. lanceolate* flowers 70% ethanol extract; CLF2: *C. lanceolate* flowers water extract. AA: ascorbic acid. All values are mean ± SD. Means with different letters in the same column are significantly different at *p* < 0.05 by *t*-test (for TPC and TFC yields) and Tukey’s test (for ABTS^+^ and DPPH).

**Table 3 antioxidants-10-00951-t003:** Correlation analysis of antioxidant activities with phenolic compound contents.

Factors	ABTS	DPPH	Chlorogenic	Quercetin	Caffeic Acid	Cinnamic Acid	ROS
ABTS	1.000	0.932 **	−0.994 **	−0.994 **	−0.996 **	−0.976 **	0.925 **
DPPH		1.000	−0.893 *	−0890 *	−0.900 *	−0.855 *	0.888 *
Chlorogenic acid			1.000	1.000 **	1.000 **	0.979 **	−0.914 *
Quercetin				1.000	1.000 **	0.983 **	−0.906 *
Caffeic acid					1.000	0.981 **	−0.913 *
Cinnamic acid						1.000	−0.842 *
ROS							1.000

Factors: ABTS and DPPH were analyzed according to the IC_50_ value; TP, total contents of phenolic compounds; ROS in PC12 cells, intracellular reactive oxygen species. Significance was determined by Pearson’s correlation coefficient; * *p* < 0.05, ** *p* < 0.01.

## Data Availability

Data is contained within the article.

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
