# Peer review of "Neuroprotective Effects of Coreopsis lanceolata Flower Extract against Oxidative Stress-Induced Apoptosis in Neuronal Cells and Mice"

_antioxidants, 2021, doi:10.3390/antiox10060951_

Round 1
Reviewer 1 Report
The present study investigated the neuroprotective effects and underlying mechanisms of Coreopsis lanceolata in vitro and in vivo model
Study findings suggest that this flower (CLF) contains various bioflavonoids with bioactivities and can ameliorate the progression and the treatment of progressive neurodegenerative disease.
The methods and the results are well done. The organization of the figures is good and clear. The study is very interesting and the use of CLF can be beneficial to slow the process of aging stopping the accumulation of oxidative stress.
Reviewer 2 Report
In this work, the authors investigated a molecular basis of neuroprotective effects of bioflavonoids extracted from flowers of one of the plans (Coreopsis lanceolata). The experimental study generally contains several flaws of conceptual basis that can be summarized as below:
Methods.
- It is not clear the rationale of using PC12 cells (a hybrid cell line) instead of implementing primary neuronal cultures in order to study “neuroprotective effects”. No information is provided for the age of the cells at the time of carrying out experiments, nor if the cells were differentiated. Had the PC12 cells some processes or not? What exact treatment scheme(s) was used for these experiments?
- How was the oral treatment controlled in animals? Why had been this approach chosen over injections that provide well-controlled delivery of a drug?
In vitro results and data interpretation.
- In vitro part of the work would much more benefit from using primary neuronal cultures instead of PC12 cells. The authors have to detail (and show images) what stage the PC12 cells were at the time of samples were collected for the analysis.
- How was the viability of PC12 cells measured (line 297)? Please show the data.
- Figure 4 shows images that are quite useless since it is not possible to even see cells, nor assess the data at all. What was the profile of ROS changes, i.e. the DCF fluorescence signal? Was the ROS increase homogenous or heterogeneous across different cells? Please provide representative images of high magnification (allowing to visualize individual cells).
- There is a lack of proper control for the effect of CLF1 and CLF2 on ROS production (Figure 4b). This should be measured in control cells (without H2O2) to test the effects of vehicle and then comparing experimental groups to each corresponding control (vehicle).
- 3 and 3.4 should be combined into a single section.
- Same as above - lack of the proper control for the expression of antioxidant enzymes (Figure 5). CLF1 should be tested at different concentrations in control (no H2O2) to compare the effect of vehicle on the level of SOD2 and CAT expression.
- Please correct your statement in lines 335-336 which is in contradiction to the data shown.
- The authors stated significant changes in some mitochondrial pro-apoptotic proteins (the increased Bax expression while decreased Bcl-2) (lines 349-351); however, their data do not support this statement - statistical summary shows no effect of H2O2 on the Bax/Bcl ratio at all (Figure 6a). Please check your data and the text provided! Instead of the Bax/Bcl ratio show changes for both proteins separately.
- It is not clear what is compared to what in Figure 6b. Please make it clear.
- All Western blot examples must show the exact MW (kDa) for the quantified protein expression bands!
In vivo results and data interpretation.
- What is the efficacy of delivery of CLF1 through BBB, especially given its oral administration?
- Investigating a level of proteins in the whole brain makes little sense as the expression profiles of varied protein are much regional-specific (section 3.7).
- Figure 7 missed the quantification summary.
Reviewer 3 Report
This article show the protective effect of two different extracts of Coreopsis lanceolata against oxidative stress in neuronal cells using both, cell culture and animal model studies.
The description of the methodology used is adequate and the results are correctly shown. However, some clarifications are needed.
Major points:
1.- Sentences must be revised along the text but specially in abstract. For example, what CLF contains various bioflavonoids with bioactivities mean? (line 19). Or, CLF inhinited H2O2-induced production of intracellular reactive oxygen species (line 25). H2O2 is not ROS but produce oxidative stress. Or MPP+-induced C57Bl/6 mice (line 30). What this means?
2.- Further, authors indicate the antioxidant activity of the compounds but in general, the main activity of these compounds is to induce endogenous antioxidant enzymatic activities in cells. This consideration must be revised in all the manuscript.
3.- Authors indicate that oxidative stress is accumulated (line 45). It is better considerate the accumulation of oxidative damage and the increase of oxidative stress.
4.- In section 3.1, the description of the peaks needs to be improved. Authors use highest several times but the meaning is not clear. What "the quercetin peak was highest in CLF1 and CLF2". This not directly indicate a higher concentration as can be seen in table 1.
5.- In section 3.2 authors indicate that the phenolic extract yields of CLF1 and CLF2 were 30.85% and 8,43% respectively. In comparison with what?
6.- In section 3.3, authors indicate that the ROS levels in H2O2 treated Pc12 cells was only 1.3 times more than in controls. However, in figure 4 A, the signal of treated cells is clearly more than 1.3. I consider that these figures and date must be revised. Further, the statistic characteristics of this study are not indicated in figure 4.
7.- The correlation shown in Table 3 is not clear. Please, clafity.
8.- In figure 5 authors show the levels of catalase and SOD levels. The quantification shown do not clearly correlates with the image of the WB. For example, in H2O2 treatments without or with 50 (microg/ml) of treatment, Catalase seems to be lower than in control, but in image is similar or higher.
9.- In figure 6, 7 and 8, blots seems to be overexposed, especially with actin, Bcl-X and Bax. A lower exposition is needed.
10.- Why the levels of active caspases are so high in control cells in figure 8. Please, revise.
11.- In lines 506 and 507, authors indicate that "CLF could be considered a new antioxidant compound". This is a mixture of compounds. Please, clarify.
Minor points:
1.- In line 43, two references are not correctly shown.
2.-OS is already indicated in line 45. Remove the indication in line 46.
Round 2
Reviewer 2 Report
I could not recommend this article at its present stage. The authors simply discarded the minimal amount of required additional experiments and ignored major issues for the data analysis and presentation.
1 – As the authors specified to use undifferentiated PC12 cells, the statement about neuroprotective effects in vitro must be changed, including the title, because it appeared to not been studied for neurons in cultures!
2 – No description (nor a carton) provided for what exact treatment scheme(s) was used for in vitro experiments, as requested.
3 – All images should show scales; the authors also have to indicate all scale values (e.g. Figure 3).
3 – The authors did not explain how they measured cell viability. It must be added to the methods section to detail how this parameter was quantified – showing one single cell image does explain how viability (by what measure?) was evaluated across cell cultures.
4 – The authors did not show any single image of DCFH-DA-labelled cells at high magnification as requested to see the pattern of changes in cell fluorescence between experimental groups.
5 – The limitations of this study should be clearly stated in the discussion section, explaining issues about yet unknown efficacy of delivery of testing compounds through BBB, and approach to oral drug administration.
6 – Investigating a level of particular proteins in the whole brain makes no sense as the protein expression profiles are very regional-specific (section 3.7).
7 – The text quality requires rigorous editing.
Reviewer 3 Report
Thanks for the changes. I have no further suggestions. I have found some typos in the new text but I think you can easily correct them in the revision of the final version.
